

# The role of ambient temperature and body mass on body temperature, standard metabolic rate and evaporative water loss in southern African anurans of different habitat specialisation

Mohlamatsane Mokhatla[1,2], John Measey[1] and Ben Smit[3,4]

[1] Centre for Invasion Biology, Department of Botany and Zoology, University of Stellenbosch, Stellenbosch, South Africa
[2] Scientific Services, South African National Parks, Sedgefield, South Africa
[3] Department of Zoology and Entomology, Rhodes University, Grahamstown, South Africa
[4] Department Zoology, Nelson Mandela University, Port Elizabeth, South Africa

Corresponding author
Mohlamatsane Mokhatla,
m.mokhatla@mail.com

## ABSTRACT

Temperature and water availability are two of the most important variables affecting all aspects of an anuran's key physiological processes such as body temperature ($T_b$), evaporative water loss (EWL) and standard metabolic rate (SMR). Since anurans display pronounced sexual dimorphism, evidence suggests that these processes are further influenced by other factors such as vapour pressure deficit (VPD), sex and body mass ($M_b$). However, a limited number of studies have tested the generality of these results across a wide range of ecologically relevant ambient temperatures ($T_a$), while taking habitat use into account. Thus, the aim of this study was to investigate the role of $T_a$ on $T_b$, whole-animal EWL and whole-animal SMR in three wild caught African anuran species with different ecological specialisations: the principally aquatic African clawed frog (*Xenopus laevis*), stream-breeding common river frog (*Amietia delalandii*), and the largely terrestrial raucous toad (*Sclerophrys capensis*). Experiments were conducted at a range of test temperatures (5–35 °C, at 5 °C increments). We found that VPD better predicted rates of EWL than $T_a$ in two of the three species considered. Moreover, we found that $T_b$, whole-animal EWL and whole-animal SMR increased with increasing $T_a$, while $T_b$ increased with increasing $M_b$ in *A. delalandii* and *S. capensis* but not in *X. laevis*. Whole-animal SMR increased with increasing $M_b$ in *S. capensis* only. We did not find any significant effect of VPD, $M_b$ or sex on whole-animal EWL within species. Lastly, $M_b$ did not influence $T_b$, whole-animal SMR and EWL in the principally aquatic *X. laevis*. These results suggest that $M_b$ may not have the same effect on key physiological variables, and that the influence of $M_b$ may also depend on the species ecological specialisation. Thus, the generality of $M_b$ as an important factor should be taken in the context of both physiology and species habitat specialisation.

## INTRODUCTION

Water and temperature are the two most important ecological determinants of species distribution patterns (*Buckley & Jetz, 2007*; *Qian et al., 2007*; *Riddell et al., 2017*) through their influence on environmental energy availability (*Currie, 1991*; *Chown et al., 2003*). At a landscape level, they influence species abundance as well as activity patterns (*Dabés et al., 2012*; *Peterman & Semlitsch, 2014*). At the physiological level, animals constantly need to maintain a positive energy and water balance in order to meet their short and long-term energy requirements for growth, development and reproduction (*McNab, 2002*). In contrast to endotherms (most birds and mammals; see *Geiser, 1998*), regulation of body temperature ($T_b$) of ectotherms is external and has been found to be highly correlated with ambient temperature ($T_a$: *Brattstrom, 1979*; *Huey, 1991*; *Angilletta, Niewiarowski & Navas, 2002*). In vertebrate ectotherms (fish, amphibians and reptiles), the regulation of $T_b$ is maintained largely through behavioural means by selecting suitable microclimates (*Lillywhite, 1970*; *Brattstrom, 1979*; *Wilson, James & Johnston, 2000*; *Seebacher & Alford, 2002*; *Köhler et al., 2011*; *Herrel & Bonneaud, 2012*; *Herrel, Vasilopoulou-Kampitsi & Bonneaud, 2014*). Thus, given the diverse array of life-history traits within anurans and associated microclimates we might expect a variation in experienced $T_b$, even within the same biogeographic setting.

The maintenance of optimal $T_b$, through the selection of suitable microclimate sites, determines how ectotherms interact with their environment. Subsequently, several processes vital for survival such as food assimilation rates, performance and ultimately fitness, are all temperature dependent (*Huey & Stevenson, 1979*; *Huey & Kingsolver, 1989*; *Angilletta, 2001*; *Angilletta, Niewiarowski & Navas, 2002*; *Seebacher & Franklin, 2005*; *Buckley, Hurlbert & Jetz, 2012*). Furthermore, $T_b$ is crucial in determining key physiological processes such as evaporative water loss (EWL) and standard metabolic rates (SMR) in vertebrates (*Huey, 1991*); such that both EWL and SMR increase with an increase in temperature (*Gillooly et al., 2001*). This is mainly because higher temperatures generally increase enzyme reaction rates (*Gillooly et al., 2001*; *Brown et al., 2004*).

These coupled increases in SMR and EWL are known to be influenced by body mass ($M_b$) such that large bodied individuals have relatively low mass-specific rates (*Gillooly et al., 2001*) of metabolism and water loss, irrespective of taxon (*Tracy, Christian & Tracy, 2010*). Moreover, rate of heat loss and gains depend on $M_b$ in anurans (*Carey, 1978*). In addition to the role of $M_b$ in determining rates of EWL and SMR, several studies suggest that habitat use of an organism can drive EWL requirements (*Thorson, 1955*). Evaporative water loss is particularly pronounced in amphibians because most have moist, highly permeable skin (*Spotila & Berman, 1976*; *Shoemaker & Nagy, 1977*). Indeed, most amphibians lack physiological adaptations to regulate water loss; thus suggesting that the rate at which some amphibians lose water is similar to an open water-body of a similar size (*Spotila & Berman, 1976*; *Wygoda, 1984*; *Tracy et al., 2007* although see *Dohm et al., 2001*; *Burggren & Vitalis, 2005*).

Evidence suggests that amphibians occupying different ecological niches show pronounced differences in rates of EWL with arboreal frogs showing comparatively
reduced rates of cutaneous water loss compared to non-arboreal groups (e.g. terrestrial and aquatic groups: *Wygoda, 1984*; *Wygoda & Garman, 1993*; *Young et al., 2005*, *2006*). In some instances, anurans achieve this by covering their bodies with a water resistant waxy secretion (see *Barbeau & Lillywhite, 2005*; *Gomez et al., 2006*), while maintaining $T_b$ above $T_a$ which increases passive heat loss, whilst reducing evaporative heat loss demands (*Wygoda & Williams, 1991*). In contrast, evidence of an ecologically mediated pattern of SMR in ectotherms seems to be lacking since differences in SMR have largely been linked to differences in species activity patterns (*Clarke & Johnston, 1999*).

Understanding the effect of temperature on key physiological traits in different anuran species (with different ecological specialisations) will enable us to better understand how changes in climate will affect this threatened vertebrate group (*Pimm et al., 2014*). They may also provide us with a snapshot of how physiological differences drive species-specific responses to climate change. Climate change is expected to be more pronounced in sub-Saharan Africa with increased drying, particularly in the winter rainfall region and variable rainfall regimen across the region (*Giannini et al., 2008*; *Engelbrecht, Engelbrecht & Dyson, 2013*), further placing amphibians of this region at high risk of extinction (*Hof et al., 2011*; *Foden et al., 2013*; *Garcia et al., 2014*; *Mokhatla, Rödder & Measey, 2015*). Southern Africa has a diverse anuran fauna occupying many different habitat types (*Alexander et al., 2004*), and we may expect variation in the way each species will respond to environmental challenges in their respective environments (*Loveridge, 1976*).

In this study, we determine how $T_a$ affects $T_b$, EWL and SMR of three different temperate African anuran species: (i) the principally aquatic African clawed frog (*Xenopus laevis*), (ii) stream-breeding common river frog (*Amietia delalandii*), and (iii) principally terrestrial raucous toad (*Sclerophrys capensis*). We tested how the variation in $T_a$ affects $T_b$, whole-animal EWL and SMR in these three anuran species. We expect that differences in species' response are associated with differences in ecological specialisation. Specifically, we expect aquatic species to show higher EWL compared to terrestrial and semi-aquatic species, as conditions are buffered in aquatic environments (excluding biotic interactions), which may lead to reduced potential for evolutionary adaptation (*Toledo & Jared, 1993*). Because of the negative relationship between water loss and activity patters (*Peterman & Semlitsch, 2014*), we hypothesise that terrestrial-adapted species would show reduced metabolic and water loss rates as an adaptation to terrestrial life (*Wygoda, 1984*). Likewise, species with intermediate life-histories (closely associated with water but not living in it), would exhibit an intermediary status. We expected that all species will maintain $T_b$ closer to $T_a$, except at higher $T_a$, where frogs were using evaporative cooling.

## METHODS

### Study species

We chose three sympatric anuran species with different habitat specialisations, based on their modes of egg deposition and development (*Mercurio, Böhme & Streit, 2009*). The pipid, *X. laevis*, inhabits and breeds in permanent water bodies. This species is usually

referred to as permanently aquatic because it possesses several key physiological adaptations (e.g. lateral line system, webbed hind feet etc.), suitable for an aquatic life style (see *Measey, 2004*). However, evidence also suggests that under severe drying conditions of permanent ponds, *X. laevis* frequently disperse overland (*Measey, 2016*). *A. delalandii* (family: Pyxicephalidae—previously known as *A. angolensis* and *A. quecketti*) breeds in and inhabits streams and flowing rivers. Adults are usually encountered on the water edge and on rocks along streams but are seldom encountered away from water bodies (*Channing, 2004*). *S. capensis* (previously known as *Bufo rangeri* and later *Amietophrynus rangeri*) is a member of the toad family Bufonidae. These toads generally breed in shallow, temporary water bodies and adult toads are adapted to a terrestrial mode of life (*Cunningham, 2004*).

Adult individuals of our three target species were caught around the area of Port Elizabeth, South Africa (Table 1). Experiments were undertaken under the animal research ethics clearance permit number A13-SCI-ZOO-007 issued by the Research Ethics Committee (Animal) at the Nelson Mandela University. Animals were collected under permit number CRO41/14CR, issued by the Department of Economic Development, Environmental Affairs and Tourism, Eastern Cape Province. We caught between 20 and 30 individuals (of varying but overlapping interspecific body size classes) excluding juveniles and gravid females. We attempted to maintain $M_b$ comparable among species as much as possible, although we found this difficult for *A. delalandii*. We included both male and female individuals to account for variation that might be a result of sexual dimorphism. Sexual dimorphism is particularly pronounced in anurans, with females generally larger than males (see Table 1 for mass differences, with *X. laevis* showing the largest differences between males and females). Animals were kept for a maximum period of 2–4 months in the lab under environmentally enriched conditions (see below), until all experiments were completed. After completing the experiments, animals were returned to their respective sites of capture.

*Amietia delalandii* were maintained in 110 L plastic boxes, with sand and small logs for cover, at low densities (five individuals per box). *S. capensis* were kept in terraria made from paddling pools ($d \times h$: 2.16 × 0.45 m) with sand, water and small logs and bark to provide cover. *X. laevis* were kept in a freshwater tank ($l \times b \times h$: 3.55 × 0.9 × 0.63 m) with stacked-bricks and stones to provide adequate cover and fed a diet of ox-heart. Both *A. delalandii* and *S. capensis* were fed mealworms and crickets, dusted with calcium (ReptiCalcium; Zoo Med Laboratories Inc, San Luis Obispo, CA, USA). All species had food available ad libitum, and the holding rooms were maintained at 20 °C, on a 12:12 photoperiod. Atmospheric air was circulated to maintain a constant temperature in the holding rooms, thus ambient humidity levels fluctuated with outside air. Following *Hillman et al. (2009)*, feeding of individuals ceased 3 days prior to experiments to ensure that individuals were post-absorptive during experimental sessions. Water was provided throughout the duration of the study (including periods when food was withheld) to prevent dehydration stress. We further ensured that the terraria in which we held *S. capensis* and *A. delalandii*, were sprayed with water every 3 days to dampen the sand and

**Table 1 Mean body sizes.** Sample size, body mass (g) (mean ± SD) and mean percentage body mass loss (mean ± SD) of individuals after 2 h of each experimental trial.

| Species | Male | | Female | | % $M_b$ loss after 2 h |
|---------|------|------|--------|------|------------------------|
| | N | Mass (g) | N | Mass (g) | |
| *Amietia delalandii* | 15 | 7.65 ± 1.61 | 9 | 15.98 ± 6.03 | 5.2 ± 2.85 |
| *Sclerophrys capensis* | 10 | 50.55 ± 8.91 | 14 | 82.58 ± 26.26 | 3.91 ± 3.84 |
| *Xenopus laevis* | 11 | 29.87 ± 10.52 | 22 | 70.75 ± 30.55 | 3.21 ± 1.66 |

we provided a bowl with water daily where amphibians could rehydrate. Mass was recorded to the nearest ±0.01 g before and after each experimental run.

## Gas exchange measurements

Standard metabolic rates and EWL measurements were conducted on the three anuran species following *Lighton (2008)* and *Steyermark et al. (2005)*, also see *Gomes et al. (2004)* for a range of respirometry methods used, particularly for ectotherms. We used an open-flow respirometry system operated on a push through mechanism on post-absorptive, non-reproductive individuals, at rest (*Sinclair et al., 2013*). Experiments were conducted at temperatures ranging from 5 to 35 °C at 5 °C intervals (*Dunlap, 1971*). The order of experimental temperature runs was randomised to reduce the effects of experimental acclimation to any directional shift in temperature. Eight individuals of each species (four males: four females) were randomly selected for trials at each temperature and a single trial was conducted on each individual per temperature.

Prior to each experimental session, we recorded $T_b$ (using a Fluke 80PK-1 probe, Type K thermocouple −40 to 260 °C, to the nearest 0.1 °C), then patted frogs dry to remove excess water from the skin and we then recorded $M_b$ to the nearest 0.01 g. All experiments were conducted between 07:00 and 18:00 h, always within the light cycle of the 12:12 photoperiod, when animals were less likely to be active (*Gomes et al., 2004*). Frogs were placed individually in suitably-sized air tight glass metabolic chambers of three sizes: 341 mL for small-, 476 mL for medium and 978 mL for large frogs, depending on the size of the individual. A similar approach was followed by *Young et al. (2005)* in order to minimise large amounts of unoccupied spaces inside the metabolic chamber (see also *Gatten, 1987*; *Gomes et al., 2004*). We found that individual frogs were agitated when they experienced respirometry procedures for the first time. Prior to the respirometry experiments, we performed a training session on each individual frog by placing them inside metabolic chambers for 15 min at 20 °C. Frogs were noticeably quiescent during subsequent respirometry runs.

A 0.5 cm layer of mineral oil was added to each chamber to prevent evaporation of excreted materials. Inside the chamber, a frog was placed on a plastic mesh platform (with sufficiently large holes for faeces to fall through), suspended at least two cm above the oil layer (*Smit & McKechnie, 2010*). Air temperatures inside the metabolic chamber were recorded using a thermocouple probe (Fluke 54II*B*; Fluke Corporation, Everett,

Washington, D.C., USA) that was inserted inside the chamber. Once the animal was placed in the chamber, we used '*cling-wrap*' (GLAD, Johannesburg, South Africa) before sealing with a glass lid. After placing the lid, we placed *Prestik*, '*Blu-Tac*' type material (Bostik, Cape Town, South Africa) around the lid of the metabolic chamber to minimise air leaks. Two metabolic chambers, one containing an animal, and the other an empty reference chamber (serving as a chamber to determine baseline levels) were placed in a custom-made environmental chamber made from a 100 L cooler box with the interior lined with copper tubing. Baseline levels were recorded for 30 min before each trial (*Smit & McKechnie, 2010*; *Van de Ven, Mzilikazi & McKechnie, 2013*). The temperature inside the cooler box was controlled by pumping temperature-controlled water through the copper tubing using a circulating water bath (FRB22D; Lasec, Cape Town, South Africa; see *Van de Ven, Mzilikazi & McKechnie, 2013*). A small fan was used to ensure air circulation inside the environmental chamber.

During SMR and EWL measurements, we used Mass Flow System pumps (MFS-2; Sable Systems, Las Vegas, NV, USA) to pump atmospheric air scrubbed of water vapour (using a Drierite column (98% $CaSO_4$, 2% $CoCl_2$, Sigma-Aldrich, Darmstadt, Germany) at a flow-rates of 100–600 mL $min^{-1}$, through the metabolic chambers. We calibrated the MFS-2 pumps using a flow-bubble meter (calibrated flow-rates were used in subsequent equations, see below). We scrubbed the air of water vapour to have better control of ambient humidity levels in the respirometry chamber (actual vapour pressures recorded for atmospheric air in Port Elizabeth varied greatly among days during our study period). Air from the metabolic chambers was sequentially sub-sampled, using Subsampler (SS3; Sable Systems, Las Vegas, NV, USA) and a Multiplexer (V3; Sable Systems, Las Vegas, NV, USA) was programmed though Expedata (Sable Systems, Las Vegas, NV, USA) to record gas concentrations for each chamber at 20-min intervals, recording an air sample every second. Subsampled air was first pulled through a water vapour analyser (RH-300; Sable Systems, Las Vegas, NV, USA) to measure water vapour pressure (WVP). We were mainly interested in measuring total EWL, not different components of water loss such as boundary layer and cutaneous resistance; hence we did not use agar models in our approach (*Buttemer, 1990*). Air samples then passed through a carbon dioxide analyser (Ca-10a; Sable Systems, Las Vegas, NV, USA) and finally through to an oxygen analyser (Fc-10a; Sable Systems, Las Vegas, NV, USA). The behaviour of the frogs was monitored during the trials using live video feed for the duration of the trial. The thermocouple probe was used to measure $T_{air}$ (air temperature inside the metabolic chamber). During a trial, frogs experienced one controlled temperature at a time. A trial was considered completed when the water vapour pressure and temperature trace was stable for 20 min or if the animal appeared too distressed to continue with measurements. Although we acknowledge that different sized frogs show differences in cooling rates (*Wygoda, 1988b*, *1989*), we assume that the final 20 min interval of stable vapour pressure and temperature trace suggest that at this point, each individuals' $T_b$ (irrespective of size) had reached equilibrium with the desired test temperature. Trials did not last longer than 2 h (following *Dunlap, 1971*). After each trial, we removed the frog from the chamber, recorded cloacal temperature ($T_b$) within 30 s of removal (using a Fluke 80PK-1 probe, Type K

thermocouple −40 to 260 °C) and the final $M_b$. Species lost on average between 3% and 5% of their initial $M_b$ (see Table 1). This loss in $M_b$ was comparable to other studies looking at water loss and metabolic rates in anurans (see *Steyermark et al., 2005*). In this study, it was concluded that the average loss of 5% in $M_b$ suggested that the animals remained adequately hydrated throughout the experimental trial (*Steyermark et al., 2005*). After each trial, frogs were individually placed in temporary holding facilities. *X. laevis* were kept in 20 L buckets, half-filled with water at 20 °C and fed ox heart. Both *S. capensis* and *A. delalandii* were kept in a 0.5 L plastic container lined with a wet lab paper and were fed mealworms and crickets, respectively. After each trial, individuals were eligible for selection for another trial run only after 3 days.

## Data extraction

Once all experiments were complete, we used Expedata software to extract oxygen, carbon dioxide and WVP traces from data files. We selected the most stable 20 min trace in each run, when the animal was at rest. Standard metabolic rates can either be calculated using the rate of oxygen consumption or carbon dioxide production rates (*Gatten, Miller & Full, 1992*; *Lighton, 2008*; *Withers, 2001*). We used rates of oxygen consumption ($VO_2$) at each temperature following *Gomes et al. (2004)*. Moreover, flow rate was calibrated using a flow-bubble meter. To determine the rates of EWL, we converted rates of WVP to water vapour density, which we subsequently converted to rates of EWL (see *Lighton, 2008*). Lastly, we calculated saturation WVP at each test temperature to determine vapour pressure deficit (VPD) as recent evidence suggests that VPD directly drives rates of EWL in amphibians (*Riddell & Sears, 2015*). We calculated VPD by estimating the saturation vapour pressure at each air temperature we studied from using the equations in *Campbell & Norman (1998)*. We converted the absolute WVP in the animal chamber to kPa and calculated the VPD as the difference between saturation vapour pressure and absolute WVP. However, our experimental design did not allow us to test for VPD directly (e.g. scrubbing water from incurrent air) but we tested if VPD was a better predictor of EWL in our test species (see Table S1). We found that VPD was a better predictor of EWL in both *A. delalandii* and *S. capensis* but not in *X. laevis*.

## Statistical analysis

To determine the effect of $T_a$ on $T_b$, EWL and SMR of anurans, we used linear mixed-effect models with the R package 'nlme' (*Pinheiro et al., 2014*) due to repeated measurements on individuals. We also included $M_b$ and sex as fixed effects in models. Within ectotherm, $T_b$ is highly correlated to $T_a$ and we thus ran a repeated measures correlation using the R package 'rmcorr' (*Bakdash & Marusich, 2017a*, *2017b*). We found a strong, significant positive correlation between $T_a$ and $T_b$ ($r_{(96)} = 0.9917$; 95% CI [0.988–0.994], $P < 0.001$) for all three species (see Fig. 1). We included sex in the model because of the pronounced sexual dimorphism in anurans. Furthermore, we wanted to determine if species differed in their responses to changes in $T_a$. However, this analysis was restricted to species with comparable $M_b$ (i.e. *S. capensis* and *X. laevis*), because it is known to be a significant contributor to observed physiological difference. We also ran a repeated measure

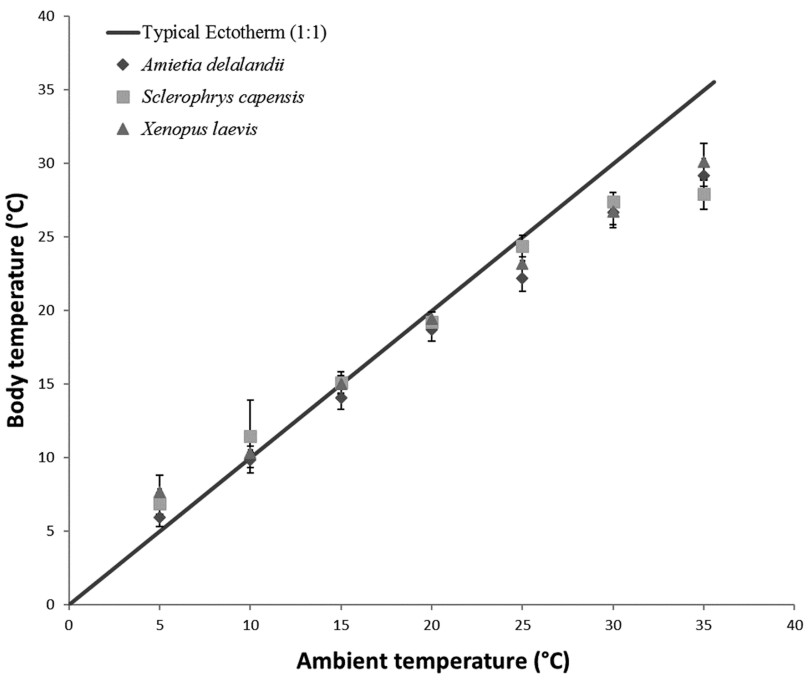

**Figure 1 Increasing body temperature as a function of ambient temperature in African anurans.** Relationship between $T_a$ and $T_b$ of different functional groups over a range of different test ambient temperatures ($T_a$). The solid line indicates $Y = X$ relationship representing a typical 1:1 $T_a$ vs $T_b$ relationship depicting an amphibian incapable of regulating $T_b$ through physiological or behavioural means. This highlights how $T_b$ deviates from $T_a$ particularly at low and high $T_a$.

correlation between flow rate and EWL to determine if the variation in flow rate had any effect on EWL. We found a non-significant relationship ($r_{(97)} = 0.157$, 95% CI [−0.044 to 0.345], $P = 0.122$) between the two variables and thus did not include flow rate as a fixed variable in subsequent models. We determined the proportion of the variance explained by the model (coefficient of determination $R^2$ for mixed models following a procedure by *Nakagawa & Schielzeth (2013)* and *Nakagawa, Johnson & Schielzeth (2017)*. Furthermore, we also determined the relative importance of each of the fixed variables in the model using the semi-partial $R^2$ as a measure of effect size (see *Jaeger et al., 2017*) using the 'r2glmm' package, implemented in R (*Jaeger & R Core Team, 2017*). All analysis were undertaken in R 3.4.2 (*R Development Core Team, 2017*).

## RESULTS

### African clawed frog: *Xenopus laevis*

We found that $T_b$ increased with an increase in $T_a$ ($F_{(6,23)}$ 1,050.629; $P < 0.0001$), although it did not affect sex ($F_{(1,23)}$ 3.440; $P = 0.0765$) and $M_b$ ($F_{(1,23)}$ 0.238; $P = 0.631$; see Table S2 for semi-partial $R^2$ values). Whole-animal EWL increased at high $T_a$ ($F_{(1,22)} = 16.201$; $P < 0.0001$; see also Table S2 for semi-partial $R^2$ values on the different temperature levels). We did not find any significant effect of EWL on sex ($F_{(1,23)}$ 3.056; $P = 0.094$) and $M_b$ ($F_{(1,23)}$ 0.198; $P = 0.661$). Furthermore, we found that whole-animal SMR increased with an

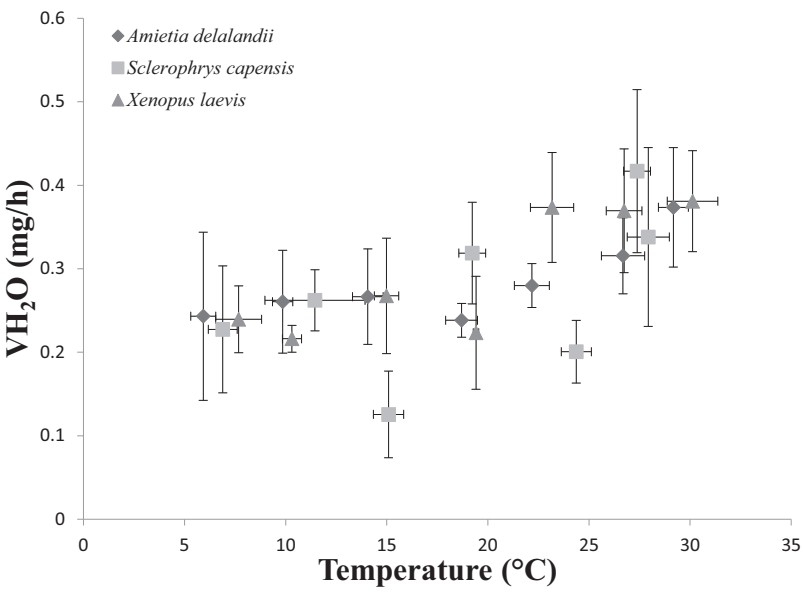

**Figure 2 Increasing rates of evaporative water loss (EWL) as a function of temperature in african anurans.** Data points represents the relationship between whole-animal EWL (as represented by $VH_2O$) and body temperature ($T_b$) at each of the test temperatures. Vertical error bars represents the variation (Standard Error: SE) in whole-animal EWL and the horizontal error bars represents the variation (SE) in body temperature.

increase in $T_a$ ($F_{(6,\ 23)} = 10.659$; $P < 0.0001$), although we did not find any significant difference among sex ($F_{(1,23)}$ 3.354; $P = 0.080$) and $M_b$ ($F_{(1,23)}$ 0.007; $P = 0.934$).

## Common river frog: *Amietia delalandii*

We found that $T_b$ increased with an increase in $T_a$ ($F_{(6,24)} = 1,374.112$; $P < 0.0001$). Furthermore, $M_b$ had a significant positive effect on $T_b$ ($F_{(1,24)} = 8.196$; $P < 0.01$; semi-partial $R^2 = 0.160$), such that $T_b$ matched $T_a$ closer in larger individuals while smaller individuals maintained $T_b$ below $T_a$. However, we did not find a significant effect of sex ($F_{(1,24)} = 0.748$; $P = 0.396$) on $T_b$. Whole-animal EWL ($F_{(6,24)} = 6.612$; $P < 0.001$, Fig. 2) and SMR ($F_{(6,24)} = 5.711$; $P < 0.0001$; Fig. 3) increased with an increase in $T_a$. We did not find a significant effect of $M_b$ in both whole-animal EWL ($F_{(1,24)} = 1.341$; $P = 0.258$) and whole-animal SMR ($F_{(1,24)} = 2.372$; $P = 0.137$), respectively (see also Table S3). Moreover, sex did not influence whole-animal EWL ($F_{(1,24)} = 0.205$; $P = 0.655$) and SMR ($F_{(1,24)} = 0.497$; $P = 0.490$).

## Raucous toad: *Sclerophrys capensis*

*Sclerophrys capensis* $T_b$ was positively correlated with $T_a$ ($F_{(1,28)} = 420.726$; $P < 0.0001$; Fig. 1). We also found that $M_b$ had a significant positive effect on $T_b$ ($F_{(1,28)} = 6.045$; $P < 0.05$, semi-partial $R^2 = 0.099$) such that $T_b$ matched $T_a$ in larger individuals while smaller individuals generally maintained $T_b$ below $T_a$, although sex did not have an effect on $T_b$ ($F_{(1,19)} = 0.020$; $P = 0.890$, semi-partial $R^2 = 0.036$). Furthermore, we found that an increase in $T_a$ lead to a significant increase in whole-animal EWL ($F_{(6,28)} = 15.055$; $P < 0.0001$; Fig. 3), although $M_b$ ($F_{(1,28)} = 3.708$; $P = 0.064$) and sex ($F_{(1,19)} = 1.190$;

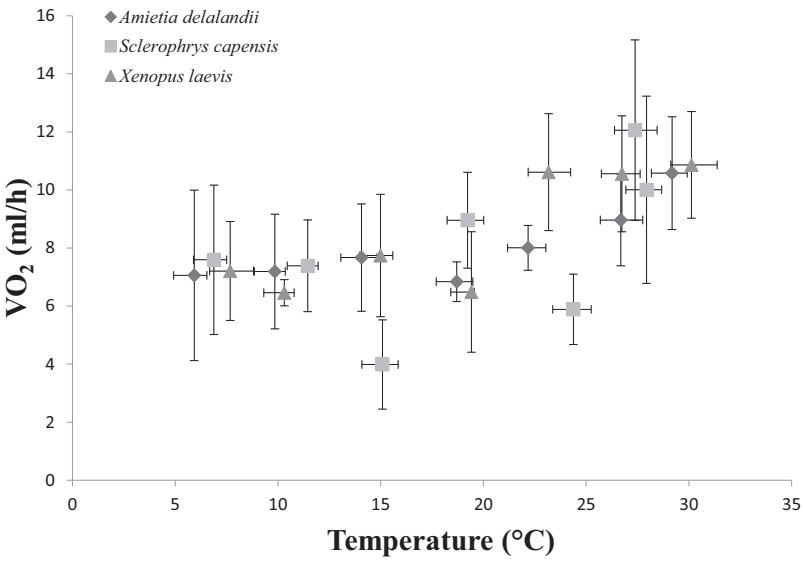

**Figure 3  Increasing standard metabolic rates (measured as oxygen consumption, VO₂) as a function of temperature in African anurans.** Data points represents the relationship between whole-animal SMR and body temperature ($T_b$) at each of the test temperatures. Vertical error bars represents the variation (Standard Error: SE) in whole-animal SMR and the horizontal error bars represents the variation (SE) in $T_b$.                                                       

$P = 0.289$) did influence whole-animal EWL (see Table S4 for semi-partial $R^2$). Whole-animal SMR also increased at high $T_a$ ($F_{(6,28)} = 11.639$; $P < 0.0001$; Fig. 3). In addition, we found that $M_b$ had a positive significant effect on whole-animal SMR ($F_{(1,28)} = 5.184$; $P < 0.05$, semi-partial $R^2 = 0.086$), although we found a non-significant sex effect ($F_{(1,19)} = 0.228$; $P = 0.638$).

# DISCUSSION

One of the greatest challenges in amphibians is that both metabolism and thermoregulation are not only coupled, but also controlled by external factors such as environmental temperature. Thus, our study aimed to assess how the variation in $T_a$ affects key physiological traits in three different African anuran species with different ecologies. We found that $T_a$ had a significant positive effect on $T_b$, whole-animal EWL and whole-animal SMR for all species, irrespective of their ecological niche. Secondly, we found that $M_b$ influenced $T_b$, in *A. delalandii* and *S. capensis* but not in *X. laevis*. Furthermore, $M_b$ did not influence rates of whole-animal EWL in all three species, but we found that whole-animal SMR increased with an increase in $M_b$ only for *S. capensis*. Lastly, we did not find any difference between the sexes in $T_b$, EWL, SMR.

## Body temperature

The concept of thermal inertia suggests that although larger individuals take longer to warm up, they also take longer to cool down (*Carey, 1978*). A study looking at how $M_b$ affects $T_b$ in different toad species concluded that larger individuals had higher thermal inertia than small sized individuals (*Carey, 1978*). In addition, *Newman & Dunham (1994)* found that *Scaphiopus couchii* toadlets that metamorphosed at larger sizes

took longer to reach critical dehydration levels compared to small-sized toadlets. Collectively, these results highlight the importance of $M_b$ as a significant factor influencing key physiological processes such as $T_b$ through its effect on rates of heating and cooling (*Wygoda, 1988a*) particularly in vertebrate ectotherms. In this study, we found that changes in $M_b$ had a significant effect of $T_b$ in *A. delalandii* and *S. capensis*. These results suggest that the effect of $M_b$ on heat flux may be more beneficial for species spending a large proportion of their time on land, where changes in temperature are more pronounced and may be sudden.

Our results seem to suggest that although $M_b$ is a key factor affecting $T_b$ in amphibians, this may not be the case across a broad range of available environmental temperatures and species ecologies. Furthermore, we found no significant relationship between $M_b$ and $T_b$ in *X. laevis* despite the species showing the largest difference in sexual dimorphism (see Table 1). *Wygoda (1988a)* suggested that prolonged cooling may be adaptive as it assists anurans to maintain a higher $T_b$ that is essential for performance under decreasing temperature conditions. Hence amphibians are expected to have larger body sizes in more temperate, cooler environments (*Ashton, 2002*) and with altitude (*Measey & Van Dongen, 2006*), although the generality of this assertion has been challenged, particularly in largely aquatic urodeles (*Olalla-Tárraga & Rodríguez, 2007*) and in anurans (*Gouveia et al., 2019*).

## Evaporative water loss

We found that whole-animal EWL increased with an increase in $T_a$ for all our species. We expected this result because amphibians use EWL to reduce heat gain, and water loss increases with an increase in metabolic rates (*Hillman et al., 2009*). Although our study was not designed to specifically test for the effect of VPD on EWL, we did show that $T_a$ was strongly related to EWL while accounting for VPD, suggesting that elevations in EWL play a role in evaporative cooling in two of the three species considered (see Table S1). We also wanted to determine whether different species differ in their ability to regulate water loss (*Wygoda, 1984*; *Young et al., 2005*). Both *Wygoda (1984)* and *Young et al. (2005)* concluded that species that adopt an arboreal lifestyle show significantly reduced levels of water loss. In addition to arboreal lifestyle, *Tracy, Christian & Tracy (2010)* concluded that both high cutaneous resistance to water loss ($R_c$), and larger $M_b$ were important in reducing desiccation time in amphibians, although high $R_c$ may not be beneficial to large-bodied frogs for thermoregulation (due to possibility of overheating at high $T_a$). In this study however, $M_b$ and sex did not have an influence on rates of whole-animal EWL in the three-species considered. While studying the relationship between desiccation tolerance and body size, *Schmid (1965)*, found no relationship between the two variables despite suggestion that small sized frogs' loose water more readily as compared to larger ones (*Heatwole et al., 1969*). Furthermore, there seems to be a difference in the onset of evaporative cooling such that species with high $R_c$ or atypical frog species only employing EWL at high $T_a$ compared low and moderate $R_c$ or typical frogs (*Tracy et al., 2008*). Species with high $R_c$ are reported to have reduced EWL and have been observed to

increase their $T_b$ above ambient as an adaptation to terrestrial habitats (*Buttemer, 1990*; *Tracy & Christian, 2005*).

*Tracy & Christian (2005)* concluded that species with low EWL had low variance in $T_b$ as a result of a negligible influence of evaporative cooling on thermoregulation, although this may not be beneficial, particularly at high $T_a$ because reduced skin resistance at high $T_a$ shortens desiccation time (*Tracy et al., 2008*). Coincidentally, species with the smallest variance in $T_b$ were also some of the largest (see *Tracy & Christian, 2005*). In the present study, we found that *S. capensis* had the smallest variance in $T_b$ between 25 and 35 °C (±4 °C as opposed to approximately 7 °C for both *A. delalandii* and *X. laevis*; see Fig. 1). Perhaps, maintaining small variations in $T_b$ particularly at high $T_a$ is beneficial for life on land. Despite $T_b$ and $T_a$ being strongly correlated, we found that $T_b$ was typically slightly lower than $T_a$ for all our species (4–7 °C below $T_a$, depending on the species), particularly at high $T_a$ suggesting that all species were using evaporative cooling at high $T_a$. Indeed, while studying toads, *Tracy (1978)* postulated that toads possess an ability to withstand higher $T_b$ for longer periods, provided that the skin remains moist. Although the ability to use evaporative cooling while managing water loss is important for amphibians, other factors such as the ability to rehydrate quickly and absorb water from a variety of substrates (e.g. burrowing species) may have been as important for amphibians to occupy such a variety of terrestrial habitats through the course of evolution (*Cartledge et al., 2006*; *Prates & Navas, 2009*), as well as being of significance to current invasions (*Vimercati, Davies & Measey, 2018*).

## Standard metabolic rates

Whole-animal SMR increases with an increase in $T_a$ as a result of the increase in kinetic energy and reaction rate required at high temperatures (*Brown et al., 2004*; *Clarke, 2006*; *Gillooly et al., 2001*). We expected *S. capensis* to have comparatively low rates of SMR (particularly at high $T_a$), as an adaptation to terrestrial life because: (i) actively searching for food (insect prey) is coupled with water loss in terrestrial habitats and terrestrial specialists should adopt a sit-and weight foraging strategy, although see *Pough & Taigen (1990)*, or (ii) only be active nocturnally or at low to intermediate levels of $T_a$ to reduce rates of EWL (*Peterman & Semlitsch, 2014*). Indeed, we found that *S. capensis* had significantly lower rates of metabolism at 15, 25 and 35 °C. Furthermore, we found whole-animal SMR increased with an increase in $M_b$ only in *S. capensis*. During species comparisons, we also found that increases in $M_b$ led to an increase in whole-animal EWL and SMR only at the highest tested $T_a$ (35 °C). This result is particularly interesting, suggesting that in Africa's temperate South, large bodies confer an advantage in delaying warming rates.

Evidence on the influence of ecology on SMR has not been clearly articulated in the literature as it may have been for EWL. Where there has been support, this has largely focused on comparing cold adapted and warm adapted species (*Clarke & Johnston, 1999*; *Addo-Bediako, Chown & Gaston, 2002*). While looking at insects at a global scale, *Addo-Bediako, Chown & Gaston (2002)* found support for the metabolic cold adaptation hypothesis. Furthermore, *Reinhold (1999)* found that flying and highly vocal insect species

had significantly high SMR than non-flying and less vocal insect species, respectively (but see also *Hodkinson, 2003*). However, *Clarke & Johnston (1999)* did not find evidence to support the hypothesis on teleost fish. While studying different but closely related anuran species, *Navas (1996)* found that high elevation species had both high metabolic rates and aerobic scope compared to their low elevation congeners. Lastly, *Navas (1997)* concluded that several factors contribute differently to adaptations to cold environments in anurans such that changes in physiology may be more important for nocturnal and terrestrial frog species.

Ecological specialisations usually occur as a result of adaptation to a finite set of environments encountered (*Navas, 1997*; *Poisot et al., 2011*). Our results suggest that although variation in $T_a$ is important in determining $T_b$, EWL and SMR in amphibians, not all amphibians are affected in a similar fashion. Although $M_b$ and sex (given the pronounced sexual dimorphism in anurans) have been identified as key factors affecting physiological traits in anurans, we suggest that perhaps this result should be considered in the context of each species' prevailing ecology and habitat specialisation with more emphasis on $M_b$. However, certain caveats to this assertion need to be considered. First, we acknowledge that the observed differences could have been because the species that we considered represent very divergent groups (three different anuran families) so that the differences are a function of the divergent evolutionary history as opposed to different ecologies (but see *Tracy & Christian, 2005*). Second, our set-up did not allow us to directly test for the relative importance of VPD on anurans, we did find that VPD was a better predictor of EWL as opposed to $T_b$ for two of the three species. *X. laevis* is a fully aquatic species and perhaps EWL may be somewhat independent of VPD over the test conditions we exposed the animals to in our study, compared to more terrestrial species. Although we welcome these new developments with VPD driving water loss rates in anurans, we also suggest that their generality should be tested against species habitat specialisations.

## CONCLUSIONS

We found that $T_a$ has a significant influence on key physiological traits in the three temperate African anuran species investigated such that it was, in most but not all, positively correlated with $T_b$, whole-animal EWL and SMR. Secondly, in *A. delalandii* and *S. capensis* we found that at high temperature EWL was mostly confounded by VPD, but not in *X. laevis*. We found that $M_b$ influenced $T_b$ and whole-animal SMR in *S. capensis* and only $T_b$ in *A. delalandii*. Lastly, $M_b$ was found not to impact rates of whole-animal EWL, irrespective of species ecological specialisation. Our results suggest that the significance of $M_b$ in influencing key physiological factors is not universal (at least for EWL) and should also be looked at in the context of species ecology as $M_b$ did not influence $T_b$, EWL and SMR in the principally aquatic *X. laevis* (see *Olalla-Tárraga & Rodríguez, 2007*; *Gouveia et al., 2019*). Understanding how species with different ecologies will respond to climate change is particularly important in anurans where some species may respond by altering their body sizes (*Sheridan et al., 2018*). Despite the pronounced sexual dimorphism in anurans, sex did not influence the relationship of $T_a$ with all physiological variables considered. We suggest future studies should focus on

disentangling the importance of $M_b$ in large sample sized, phylogenetically related, non-arboreal, anurans as information of how $T_a$ affects key physiological traits in these species is currently lacking. Furthermore, future work should be undertaken to understand energy demands of different species, such as stream breeding species, given the threats that they currently face (*Sodhi et al., 2008*) particularly when considering the expected effects of climate change on amphibians (*Buckley, Hurlbert & Jetz, 2012*; *Deutsch et al., 2008*).

## ACKNOWLEDGEMENTS

We would like to thank Ryan Rambaran, Magda Hawkins, Adrian Evans, Melissa Scerbo, Stephanie Martins, Werner Conradie and Bryan Reeves for their help in the field. We would also like to thank Sarah Davies and Giovanni Vimercati for fruitful discussions throughout the different stages of this manuscript. We would also like to thank both reviewers for their constructive comments that they provided. We firmly believe that their contributions have significantly improved the quality of this manuscript.

### Funding

This work was supported by the National Research Foundation (NRF) of South Africa as a PhD Innovation Bursary (No. 84855) and the Nelson Mandela University Post graduate bursary to MMM. JM also received support from DST-NRF Centre of Excellence for Invasion Biology, National Research Foundation of South Africa (NRF Grant No. 87759), and NRF incentive funding. The funders had no role in study design, data collection and analysis, decision to publish, or preparation of the manuscript.

### Grant Disclosures

The following grant information was disclosed by the authors:
National Research Foundation (NRF) of South Africa as a PhD Innovation Bursary: 84855.
Nelson Mandela University Post graduate bursary to MMM.
DST-NRF Centre of Excellence for Invasion Biology, National Research Foundation of South Africa: 87759.
NRF incentive funding.

### Competing Interests

John Measey is an Academic Editor for PeerJ. The authors declare there are no competing interests.

### Author Contributions

- Mohlamatsane Mokhatla conceived and designed the experiments, performed the experiments, analysed the data, contributed reagents/materials/analysis tools, prepared figures and/or tables, authored or reviewed drafts of the paper, approved the final draft.

- John Measey conceived and designed the experiments, contributed reagents/materials/analysis tools, prepared figures and/or tables, authored or reviewed drafts of the paper, approved the final draft.
- Ben Smit conceived and designed the experiments, performed the experiments, analysed the data, contributed reagents/materials/analysis tools, prepared figures and/or tables, authored or reviewed drafts of the paper, approved the final draft.

### Animal Ethics

The following information was supplied relating to ethical approvals (i.e. approving body and any reference numbers):

The Research Ethics Committee (Animal) of the Nelson Mandela University provided full approval for this research under the the ethics clearance permit number: A13-SCI-ZOO-007.

### Field Study Permissions

The following information was supplied relating to field study approvals (i.e. approving body and any reference numbers):

Individuals representing species of interest were collected under permit number CRO41/14CR, issued by the Department of Economic Development, Environmental Affairs and Tourism, Eastern Cape Province.

### Data Availability

The raw data is available as Supplemental Files.

### Supplemental Information

Supplemental information for this article can be found online at http://dx.doi.org/10.7717/peerj.7885#supplemental-information.

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
