# Peer review of "The role of ambient temperature and body mass on body temperature, standard metabolic rate and evaporative water loss in southern African anurans of different habitat specialisation"

_PeerJ, doi:10.7717/peerj.7885_

## Round 0.1 · original submission · Major Revisions

While I find the study very interesting in general, based on the comprehensive comments of the two reviewers I think that the manuscript needs a careful in-depth revision before it becomes acceptable in PeerJ. That said, the constructive nature of the reviewers’ comments should make it easy for you to thoroughly revise the study.

Both reviewers have raised a number of concerns regarding the methods and the experimental design as well as the main message and structure of the paper. Therefore, please provide a detailed point-by-point reply to the comments, with a clear description how and where you have addressed them, or with a well-reasoned explanation in case you have not followed the reviewers’ suggestions.

Please pay particular attention to the methodological issues raised by reviewer 1 (variation of flow rates, consideration of vapor pressure deficit, inflation of t-tests, etc.), but also those of reviewer 2.

Please also note that rephrasing the manuscript from a hypothetico-deductive style towards a more explorative storyline (reviewer 2) does not at all lower the chances of final acceptance!

I look forward to the revision!

Reviewer 1 ·

Basic reporting

The authors need to explore the consequences of the interaction between temperature, vapor pressure, and evaporative water loss (see specific comments below). Also, resistance to water loss is not discussed properly. Exploring these ideas might help hypothesis development, which was not very well developed.

Figure 2 and 3. The bubble plots are confusing and unnecessary. I would explore a simple regression. Use statistics to talk about why body size is important. The reader gets nothing from the bubbles.

Experimental design

Major criticism: The authors state that flow rates varied from 100 to 600 mL/min. This is troubling because flow rates can have a big influence on the rate of EWL. The authors might prove that this wasn’t the case by regressing flow rate against EWL in the supplemental material. Otherwise, flow rate must be included as a covariate. Can you explain why you varied flow rate so greatly? (Wind speed is calculated by multiplying flow rates by the cross sectional surface area of your chamber.)

line 196: By scrubbing water vapor from the system, you did not control anything about vapor pressure. In fact, you confounded your results with the vapor pressure deficit (VPD). The VPD is the factor that drives rates of evaporative water loss, and VPDs increase exponentially with temperature. Therefore, the results from EWL are primarily driven by the VPD, a direct consequence of warming temperatures. What we know, however, is that when controlling for different VPDs, amphibians respond independently to temperature and VPD (Riddell and Sears 2015). So controlling for these effects is critical in experimental physiology. If you do not control for it, then you cannot determine whether a physiological response of water loss is due to temperature or VPDs. It’s not a death sentence, by any means. Most scientists don’t control for it, but you cannot state that you controlled humidity because you cannot determine whether the change in water loss rate is due to temperature or VPD.

Validity of the findings

The conclusion that “ecology matters” is not adequately supported, and based on the narrative in the discussion, the authors understand why they cannot make this argument. They selected three highly divergent species; any physiological differences could have been equally due to evolutionary history as ecological pressure. Perhaps the authors can discuss the ecological and evolutionary implications in the discussion, but they certainly cannot conclude anything about the how ecology matters in this case. These conclusions must be removed from the paper, particularly without an appropriate analysis. That being said, I’m not entirely sure what the message of the manuscript will be. The authors should think more carefully about the message.

I am also concerned about their analysis comparing two species. I understand why they did not include the third species in their analysis, but including mass as a covariate in your analysis could have statistically controlled for these differences. If the authors want to compare slopes and intercepts of the relationships, they will also need to standardize covariates and center over zero. As an alternative, they could examine the residuals of physiological traits regressed against mass. Finally, their two species analysis conflicts with the prevailing literature that has determined two species analyses are inherently flawed (see Adolph and Garland 1994). So removing the third species was not necessary, and in doing so, the authors ignore important literature regarding multiple species comparisons. I would either analyze each species separately or include species as a random factor in your mixed model analysis. You can then look at the standard error of intercepts (and whether they overlap zero) to see if including species as a random effect mattered in your analysis.

In general, I have an issue with the dozens of t-tests in your results. Why not use a ANCOVA or similar hypothesis test to evaluate these differences? Then you can calculate effect sizes to look at the relative influence of all of your factors and covariates. Without some sort of effect size or measure of fit, I have no idea how important your covariates are. As you can tell, I am not a big fan of solely relying on p-values to interpret data.

Additional comments

Line-by-line

Abstract:

23: What does “available” ambient temperatures mean? Ecologically-relevant?

36: I would remove the last line about sexual dimorphism being important because you did not mention that it mattered in the results portion of your abstract. Or tell us why sex mattered in your experiments.

41: See Riddell et al 2017 for explicit evidence that water loss physiology structures the fundamental niche of amphibians.

72: but see Burggren and Vitals 2005 and Dohm et al. 2001.

76: Waxy secretions are actually quite rare. What’s more likely are differences in skin resistance to water loss due to changes in blood delivery to the skin due to perfusion or lipid composition of the skin.

78: It also feels like you just contradicted yourself. Check the logic here.

194: Variable flow rates are a potential methodological issue. See major criticism.

240: Was species a random effect? See major criticism.

254: Change ‘did differ’ to differed - more clear.

256: I think body mass didn’t influence either of these because of flow rate issues, especially water loss. Please include flow rates as a covariate.

357-359: Your expectations are missing a critical components about the role of temperature and water loss (see comment for line 196).

364: Skin resistance is not an ability. It’s simply a property of the animal’s skin, specifically the resistance of their skin to water loss. It’s analogous to resistances and conductances from circuit theory in physics.

365: The authors should remove this conclusion because they didn’t calculate resistances. Therefore they can’t make any conclusions on the relative importance of body mass on water loss physiology - at least not completely. If you calculated resistances you could get an entirely different pattern than what you are reporting. Your ability to discuss the physiology of these animals is therefore limited.

369: Remember that resistance does not necessarily indicate the rate of water loss. The rate of water loss is determined but the combination of resistance to water loss and the evaporative demand of the air. Resistance units are time per distance (i.e., seconds per centimeter is commonly used). A rate of water loss would be grams of water lost per unit of time. I would make that clear to the reader; otherwise, your language is confusing.

393: I find it hard to believe that food isn’t limiting in aquatic systems. Provide citation.

395: citation format issue.

416: or evolution. Your conclusion is arbitrary.

Reviewer 2 ·

Basic reporting

Mokhatla et al. – The role of ambient temperature and body mass on body temperature, standard metabolic rate and evaporative water loss in southern African anurans of different ecologies.

In general, I found the dataset very interesting but a bit confusing in the manner that authors showed them in the manuscript at its present format. I had to struggle to follow some of the text of the paper. For example, it is not clearly stated in a logical sense and sequence in the text the direction of what authors expected as a whole with what they reached in terms of novel findings. It is very noticeable when reading the main goal of the paper (Lines 26-27) and then reading the presentation of all the results (in several directions sometimes without connection, Lines 30-33) in the Abstract, i.e. many of the results do not answer the main hypothesis(es) of the paper in a logical sense (many factors precluding what is central and what is peripheral). In this way, the idea behind the paper became explorative rather than hypothetico-deductive, as can be observed in the Discussion section (many parts hard to show a logical sequence). Making things clear, concise and concatenated in the text may improve the comprehension of the text for a broader audience of readers as well may highlight even more the importance of the main results found by the authors. I suggest to keep the manuscript straightforward and as simple as possible, with temperature as central and other factors with potential to affect temperature changes on physiological traits (for example, with your experimental design you cannot say if it was body mass or gender the most important factor affecting physiological traits under temperature changes). My impression when reading was that authors were trying to introduce many factors to justify the study and highlight it as unique. I would be cautious about saying there are few studies (Line 22) on topics that you are exploring; I left some references below. In addition, I am afraid at its present form the study suffers from some methodological problems, which prevent the interpretation some findings. I made specific comments and suggestions below. Hope these are clear.

Abstract and Introduction with consequences in the manuscript as a whole

In their own words, authors aimed to determine the role of ambient temperature (Ta) and on body temperature (Tb), whole-animal standard metabolic rate (SMR) and whole-animal EWL (Evaporative Water Loss) in three amphibian species with differential ecologies from Africa (see Lines 26-28; I comment below on two misconceptions presented here). They expected that differences in species' response would be associated with differences in ecological specialization of each species, and body mass and sexual dimorphism would also be important factors influencing such responses. Authors specified they expected that the aquatic species would show higher SMR and EWL in comparison with the semi-aquatic species (intermediary SMR and EWL) and with the terrestrial species (reduced SMR and EWL) (Lines 94-108). What authors expected to happen in relation to the Tb of the amphibians was not clearly presented as hypothesis (at least in the initial sections of the paper – Abstract, Introduction – in which all ideas are, or should be clearly, presented). Moreover, although they described certain effects of temperature changes on Tb, SMR and EWL in some parts of the Introduction, they did not clearly state the direction of what was expected both in terms of specific or general results as a whole in regard to temperature changes; see the Abstract (Line 27: “…determine the role of Ta…”) and also the final of the Introduction (Line 94: “…determine how ambient temperature affects…”, in the Lines 98-99: “…tested how the variation in Ta affects…”, and in the Lines 99-100: “…determine whether these species differ in their responses to Ta fluctuations…”).

Likewise, it is hard to the reader to disentangle what is essential to pay attention in regard to the main hypothesis/es of the manuscript, if it was/were not clearly stated. As far as I understood, expected variations in Tb, SMR and EWL due to changes in an experimental abiotic factor (Ta) and/or in other factors (body mass, sexual dimorphism, habitat use). But see that only habitat use is mentioned at the end of the Introduction regarding to the expected results, which should answer the hypothesis/es. In this context, would body mass and gender be only factors of control for tests of temperature effects on physiological traits or would they being also tested as factors as well temperature? In addition, variation in the “whole-animal” and “mass-specific” SMR and EWL unexpectedly appear for the first time in the text at the final of the Introduction, also contributing to make the reader confused about where authors are and where they are going. At this point, I was again, with an impression that authors were trying to promote their study. It is not a problem at all, but it should be clear why whole-animal or specific-mass differed and what are the implications for that.

Thus, the introduction needs more focus on what are authors really planning to answer. I suggest the use of the points mentioned above to improve the manuscript goals and hypothesis/es in a clear and conciseness manner to provide more justification for your study. This will also help you expand upon the knowledge gap being filled. Specifically, the authors’ interpretation (and justification for carrying out this study) that “few studies have tested…taking species ecology” is not, at my point of view, suitable. Several studies may contradict this statement since many of them have tested SMR (Gatten et al. 1992 for a review), Tb (Brattstrom, 1963, Ecology), EWL (Tracy et al, 2007, Ecology; Lillywhite 2006, JEB for a review) under a set of experimental conditions, body masses, species ecology. Therefore, try to build logical hypothesis and predictions to match your results (with temperature variation as central). You may also reinforce the importance of including sexual dimorphism (this is rare in the literature) as factor influencing temperature changes on your physiological traits (at least for amphibians).

Minor (but not least important):

Lines 26-28: Change “determine” to “investigate” or “evaluate” or “assess” or “estimate” or a similar term. “Determine” is not suitable or proper in these circumstances since no one can determine animal responses other than them. In addition, different “habitat use” should be used instead of differential “ecologies”, a term that is vague in the context that was presented.

Experimental design

Methods:

Why the methods for obtaining Tb’s did not appear in the Methods section? I confess I still can not clearly understand how the authors obtained or derived the body temperatures (Tb’s). The Tb’s are show for the first time as results in the Results section, including the way that authors obtained/derived them. Moreover, if I have interpreted correctly, there is no sense what authors tried to do since the relationship between air, skin and core temperature can change a lot depending on the conditions of measurement and the range of temperatures over which you look. For example, the air or substrate at a given location does not predict the body temperature of the amphibian at that location, because it doesn’t include particular conduction with the substrate, summed radiation from across the entire substrate and the rest of the environment plus convection and so forth. Thus, I think you cannot oversimplify that relationship and then deriving Tb from Ta since air temperature interacts with the body of an amphibian to set its body temperature in a non linear or simplistic way. Also, depending on the interplay with hydration level (my comment below, Line 217), Tb may be very different from expected from a linear regression with Ta. My suggestion is to the authors delete these Tb from the paper results (see that Tb is not well contextualized in the hypotheses; I made comments above). Maybe try to explore the rates of Tb exchange (cooling and warming) and the levels of hydration (initial and final body mass measurements).

Lines 139-145: Anuran species were fed with different diet. Please explain why you used this treatment since different acclimation conditions may affect physiological traits in a differential way. Also, explain why you are confident that individuals did not get food differentially since there were more than one per plastic boxes at least for 2 species (there is no information on the third, Xenopus laevis). Different captivity conditions ("treatments") may imply in differences when analyzing physiological responses.

Lines 169-171: Please provide an explanation on why chambers with different sizes would not affect the final results when comparing rates of SMR among species.

Lines 156-223: Please separate in two sections: SMR and EWL measurements. It is very confusing at the present form.

Lines 213-216: Why did you assume that stable vapour pressure and temperature trace suggest equilibrium of Tb with test temperature? It is not logical. Thus, provide a rational explanation and, if possible, data to confirm that.

Line 217: Why did you used 2 hours for EWL measurements for all individuals independent of their body size? Do you have evidence that this time period did not caused dehydration, and then you include another factor other than temperature influencing the rates of EWL?

Minor (but not least important):

Line 138: Returning animals kept in captivity for 2-4 months to nature should be avoided. There is very little knowledge about what could be being introduced in terms of new microorganisms, pathogens, etc.

Validity of the findings

Results, Discussion, Conclusions

Since I have several concerns on the way the manuscript was organized and also that data was obtained, impact and novelty can not be entirely assessed at its present form.

Lines 300-308: Thermoregulation and Tb relationships should not be discussed without referring their interplay with hydro-regulation. Even if you mention it in some lines below at EWL sub-section, the influence of body mass on Tb and metabolism can not be decoupled from hydration state. If so, please provide rational explanation.

Lines 356-387: Why discuss cutaneous resistance and not rates of EWL if you only measured rates of EWL? Please provide rational explanation.

Line 418: “…Ta influence on..” in which way or direction? Describe it better.

Figures: I found Figs 2 and 3 very ingenious. But I can not obtain simple information as main descriptive tendencies. Would it be possible to include such information as central tendencies (mean or median) and variances (SD or SE)?

Additional comments

All my comments and suggestions are covered above.

---

## Round 0.2 · Major Revisions

Thank you very much for your revision, which has greatly improved the manuscript according to the opinion of both reviewers. However, both of them have again raised a number of points which I would like to see thoroughly addressed in another round of revisions.

In particular, I would like to ask you to carefully consider the comments made by reviewer 1 on the VPD analysis as well as the suggested solutions. But also the other comments of both reviewers deserve careful attention.

I look forward to receiving a new version of the manuscript!

Reviewer 1 ·

Basic reporting

no comment

Experimental design

I have an issue with the analysis of VPD that needs to be addressed. Please see general comments to the authors.

Validity of the findings

I have an issue with the validity of their analysis on VPD, see below.

Additional comments

Dear authors,

Thank you for the revision and the taking the time to address my comments. I appreciate the improvements, and I think the manuscript is useful and will be appreciated. But there’s one critical detail to address.

After reading the authors’ revision and inspecting their data, I think it’s imperative to make some things clear about their VPD analysis. Again, I appreciated the additional analyses in this manuscript, but the results are misleading. The authors state that VPD does not affect water loss rates, but in fact, their experimental design precludes an appropriate analysis of VPD. In the design, VPD and temperature are highly correlated. That’s unsurprising given that the authors scrubbed the airstream of water vapor. To analyze VPD, the authors binned VPDs into high and low categories within each temperature treatment. There are four issues with that: (1) there is very little variation in VPD within each temperature treatment leading to arbitrary binning of VPDs into categories, (2) VPDs in the high category are lower than VPDs in the low categories in different temperatures (and vice versa), (3) high or low VPDs are not present in some of the temperature treatments, and (4) an interaction term between VPD and each temperature treatment (since they are factors in the analysis) needs to be incorporated into the model to get an idea of VPDs effect within each temperature treatment. Given the first 3 issues, I would not recommend following the 4th recommendation.

That being said, there is a way to evaluate the relative role of temperature vs VPD, even though they are highly correlated in the authors’ dataset. I would recommend using VPD and air temperature (Tcham) as covariates in separate analyses in nlme and compete them to see which variable has the highest R2. When I did that analysis, VPD was a better predictor than temperature in 2/3 of the species (code below). For me, the statistical evidence and physical expectation of VPD’s effect on water loss rates suggests that the authors should be using VPD as a covariate, and not use temperature at all.

I have a feeling that the authors will have an aversion to that analysis because the predominant way of thinking about physiology is through the lens of temperature. So I suggest another option for the authors: remove VPD from the analysis, add a supplemental figure or analysis on VPD alone (as I provided), and indicate in the manuscript that the authors did not design the experiment to test for VPD but it’s possible it may be driving the effect of temperature. It could simply be a couple sentences in the results and discussion.

Minor comments

What are the units of VPD? And please provide a citation for the equations that you used to calculate it.

line 26: typo “and on”

line 33: change “and” to “or”

#R code
library(dplyr)
library(nlme)
library(MuMIn)

peerj <- read.csv("~/Desktop/peerj.csv")

#make subsets of data for each species
ami <- select(filter(peerj, Species == "Amietia"),c(Species,ID,Sex,Fri,Tcham,Mass_F,Mass_Log,VH2O,MVH2O,VPD,VPD_C))
scl <- select(filter(peerj, Species == "Sclerophrys"),c(Species,ID,Sex,Fri,Tcham,Mass_F,Mass_Log,VH2O,MVH2O,VPD,VPD_C))
xen <- select(filter(peerj, Species == “Xenopus"),c(Species,ID,Sex,Fri,Tcham,Mass_F,Mass_Log,VH2O,MVH2O,VPD,VPD_C))

#compare r.squared’s for each species with VPD and temperature
ami_mod <- lme(VH2O ~ VPD, random=(~1|ID), data=ami)
scl_mod <- lme(VH2O ~ VPD, random=(~1|ID), data=scl)
xen_mod <- lme(VH2O ~ VPD, random=(~1|ID), data=xen)

#results
r.squaredGLMM(ami_mod)
R2m R2c
0.3126473 0.4313033

r.squaredGLMM(scl_mod)
R2m R2c
0.2231434 0.2231434

r.squaredGLMM(xen_mod)
R2m R2c
0.4368583 0.4368583

ami_mod <- lme(VH2O ~ Tcham, random=(~1|ID), data=ami)
scl_mod <- lme(VH2O ~ Tcham, random=(~1|ID), data=scl)
xen_mod <- lme(VH2O ~ Tcham, random=(~1|ID), data=xen)

#results
r.squaredGLMM(ami_mod)
R2m R2c
0.2647765 0.3553159

r.squaredGLMM(scl_mod)
R2m R2c
0.214727 0.214727

r.squaredGLMM(xen_mod)
R2m R2c
0.4501313 0.4501313

Reviewer 2 ·

Basic reporting

This revised manuscript is much improved over the previous version. It is more readable than before and seems that authors dealt with almost all the methodological issues raised before by the reviewers. However, the writing still needs some attention to clarity, mainly in the Discussion section. With some further work, I find that it will be a good contribution to the knowledge of the area.

Experimental design

Ok with almost all replies/clarifications from the authors. But see my new comments below at “Authors replies and my new comments” and also at "Line-by-line".

Validity of the findings

I could not follow the rationale of not including interactions among variables in the models. Please also deal with the line-by-line comments below to clarify the new issues raised at the present version of the MS.

Additional comments

Line-by-line:

Line 26: “…the role of Ta and on Tb, whole-animal EWL and whole-animal SMR…” to keep this sequence in which the results are presented below and also in other sections. Also keep it throughout the text in order to facilitate to the readers.

Line 32: “Tb” instead of “body temperature”.

Lines 66-67: Gillooly et al. 2001 as citation for “low mass-specific rates of …water loss”?? If not, please provide a satisfactory reference for water loss issue.

Line 69: ecology is a vague term here. Following the references cited in this passage, it should be “habitat use” or something in this line.

Line 73: typo? “,” instead of “;”

Line 79: “other groups”. Which ones? Maybe something in these lines: “compared to other groups, such as terrestrial and aquatic ones”.

Lines 82-83: increases passive heat loss? Or decreases? Increasing skin resistance and consequently reducing EWL would reduce heat loss, so that overheating is a problem for such animals at high temperatures.

Line 84: "skin resistance to EWL" instead of "skin water resistance". To keep it similar with usual terms in the literature of the area.

Lines 84-85: “due to changes in blood delivery to the skin following its lipids composition.” Blood delivery of what? Maybe reword this passage in order to clarify it.

Line 100: “affects Tb, EWL and SMR” instead of “affects SMR, EWL and Tb”.

Line 101: Delete: "We selected species depending on their apparent dependency on water". Thus, the new version: "…African anuran species: the principally-aquatic African..."

Lines 104-106: I still cannot get your point here (and I commented on this in the last revision): "We tested how the variation in Ta affects (...) Next, we determine whether these species differ in their responses to Ta fluctuations."
What is the difference between these questions? It is not clear to the readers and authors should be more specific. In addition, if the term "determine" is kept after the modification, change it to another similar and more appropriate word.

Lines 106-107: “We expect that differences in species’ response are associated with differences in ecological specialisation.” I suggest that you follow the sequence that you presented your physiological variables: Tb, EWL, SMR. So: “…specialisation. We expected Tb increasing with Ta, but also Tb to be influenced by Mb, such that larger individuals would have lower Tb, specifically at higher Ta. We also expect aquatic species to show higher EWL…”.

Lines 107-110: It is a bit inconsistent once you describe in your methods (lines 121-122) that the aquatic species (X. laevis) frequently disperse overland. So why would you expect it to show higher EWL compared the semi-aquatic species? In other words, why conditions that may lead to reduced potential for evolutionary adaptation (of water balance traits?) are buffered in aquatic environments of X. laevis and not of A. delalandii?
The problem here is the way that you described it in the text, so that it makes this inconsistence evident when aligning it with your result expectations.

Line 256: I am not sure if I fully understood your data analyses. Please provide the rationale of not including interactions among variables in the models.

Line 271: Change to: “We found that changes in Tb were not related to Mb…”

Line 275-276: “sex” instead of “males and females”. Keep the terms similarly throughout the section.

Line 277: “had a significant effect of…”. What does it mean? Would it be(?): “whole-animal SMR increased with Ta”.

Line 281: “We found that changes in Ta had a significant effect on Tb”. Would not be better to describe the direction of this effect instead of just reporting that there was an effect (that the readers may be still not familiar with)? Whenever possible, provide this kind of information (direction) to the readers.

Line 353: “…aquatic urodeles (Olalla-Tárraga & Rodríguez, 2007) and in anurans (Gouveia et al., 2019)”.
https://www.amnat.org/an/newpapers/JanGouveia.html; https://dx.doi.org/10.1086/700833.

Line 356: when I go to see the Figure 2 to check it, this not apply for Sclerophys (15C and 25C are lower than 5, 10, 20C for example). This should be well explained in the text here.

Line 361: change “determine” to “investigate”.

Line 365-366: Line 365-366: The definition of Rc is not the “ability” to reduce rates of water loss. This reads awkward (and should be modified or excluded) since Rc is a value of mass (water) transference through a physical barrier, which is calculated from EWL rates.

Line 367: Maybe should be: "although high Rc may not be beneficial to large-bodied frogs for thermoregulation (due to possibility of overheating at high Ta's)". Since some of the readers may not be familiar with the issue.

Line 368-369: "However, in this study we did not find any significant effect of Mb, VPD and sex on rates of whole-animal EWL in the three-species considered".
First, it seems not presented in logical sequence since you were discussing Mb and Rc effects in the sentences above. So, VPD and sex suddenly appeared in a mixed way at this point of the discussion. I suggest you modify it. Second, "Mb... did influence whole-animal EWL" (Lines 301-302), and so this needs clarification.

Lines 369-371: I could not understand this sentence. This should be rewritten for clarity. “Furthermore, there seems to be a difference in the onset of evaporative cooling with high Rc or atypical frog species only employing EWL at high ambient temperatures compared low and moderate Rc or typical frogs”.

Lines 372-375: Again. I could not follow the rationale of your sequence in this sentence since in the last two sentences you were discussing about relationships among EWL, Rc and thermoregulation. Then, you conclude the paragraph with "and in this study we found that Tb is influenced by Mb in both S. capensis and A. delalandii". Clarify why you included Mb at this point of the text or change it accordingly.

Lines 376-378: Although you "expected that S. capensis would be capable of maintaining Tb close to Ta (based on its terrestrial life style), better than the other two largely aquatic species", it does not appear in any other part of the text. Thus, you could include it at the final of the Introduction section to previously communicate your expectations to the readers.

Lines 383-386: I suggest avoiding hyperbolical expression as "largely follows" since your dataset contains a limited range of species and habitat uses in comparison to Young et al 2005. Also, in the specific case of R. marina, I do not understand why you are not discussing the role of Mb component at this part of the text, since that is why this species is considered “atypical” (large body-size decreases the rates of EWL). Thus, this paragraph needs some work to connect variables of the problems you are discussing as well your own findings.

Lines 388-394: Very good discussion and connection of ideas. So, it is an additional motivation to improve the text above.

Lines 396-409: That is the only paragraph discussing SMR since the other one at this same sub-section (Lines 410-423) seems not a part of it. Thus, SMR section needs some further work to align with other sub-sections above (Tb and EWL).

Lines 405-406: It was what I would expect for both SMR and EWL since this species presents the largest body size in comparison to the other ones. For this kind of comparison (which led to the conclusion of the paragraph), I think it was more appropriate to compare specific values of EWL and SMR (V of H2O or O2 per gram or cm2 of animal per time). I am not sure if you can freely conclude this last idea with absolute values.

Lines 410-423: This in the context of habitat use and not of SMR, thus should be shorten (since it repeats some points later as ecology, habitat specialisation) to be incorporated to the conclusion section.

Lines 413-416: I commented on this in the last revision that this one of the very interesting finding of your study.
Line 423: This sentence closes the paragraph being very unspecific in my point of view. Ecology of what? Habitat use? Habit? Complete it according to your theoretical variables.

Line 426: Change to : "investigated such that it was, in most but not all cases, positively correlated with..”

Lines 429-431: Mb (allometry) is important when analysing all physiological processes. So I could not follow what was the point that you highlight here since Mb must be controlled before analysing changes in physiological traits. The same apply to the Lines 434-437. Please modify the text accordingly or highlight better what was your point in these paragraphs when discussing the role of Mb.

Line 431: “(Olalla-Tárraga & Rodriguez, 2007; Gouveia et al, 2019)”

Figures 2 and 3. Is it correct for 30C and 35? In my pdf version it seems they are overlapped (EWL and SMR data). Also, it seems that the terrestrial species show higher rates of EWL than the aquatic ones at 20C and 30C. Why?


Authors replies and my new comments.

Authors: Incorrect. The reviewer may have missed this point as it was outlined in line 217-220. See also lines line 178 – 180.
My new comment: Actually, it was not clear in the last version and I can see that you modified the text to achieve clarity.


Authors: We are not sure what the reviewer identifies as the problem here. It is standard procedure in respirometry to adjust chamber size to animal size, and subsequently adjust flow rates to obtain acceptable signals of gasses measured. The animal should be able to rest comfortably in the chamber and turn around, but not walk and jump around. We adjusted flow rate with chamber size aiming to maintain similar humidity and oxygen deficits in the chambers.
My new comment: Ok, but are you sure that the same apply for EWL measurements? That is why I raised this issue in the last round asking for EWL and resistance comparisons but I could not find it in the new version.

Authors: The percentage body mass loss in the 2 hr experimental period was (mean ±SE) 3. 2 1± 1.66 (Xenopus laevis), 3.91 ±3.84 % (Sclerophrys capensis and 5.2 ±2.85 % (Amietia delalandii). We have to assume the “other factor” referred to here, concerned VPD? Please see our response further above where we check that VPD did not confound our analyses as far as we could tell.
My new comment: I suggest you include all these percentages in the methods section.

---

## Round 0.3 · Minor Revisions

Thank you very much for your revision - the manuscript has, again, improved considerably. However, reviewer 2 has pointed out several issues which I kindly ask you to address carefully where appropriate.

In addition to that, I have some very minor edits which you should implement. I look forward to seeing the revised - and potentially final - version of the manuscript.

Minor edits:
24: ecologically-relevant -> ecologically relevant
97: principally-aquatic -> principally aquatic
98: principally-terrestrial-> principally terrestrial
125: three-target -> three target species
133: as result -> a result
148: check reference format - should be Hillmann et al. (2009) instead of (Hillmann et al. 2009)
173, 198: respiromtery -> respirometry
238: check reference format (see above)
281: match -> matched; maintain -> maintained
282: add "a" before "significant"
307: replace comma by "and" between "Tb" and "whole-animal"
308: delete comma after "Tb"
347: check reference format (see above)
398, 399: insects -> insect
411: After "first" I would expect a "second" somewhere
425: delete comma after "that"
426: add comma before "but"

Reviewer 1 ·

Basic reporting

I have no further comments. The article is acceptable.

Experimental design

I have no further comments. The article is acceptable.

Validity of the findings

I have no further comments. The article is acceptable.

Additional comments

I have no further comments. The article is acceptable.

Reviewer 2 ·

Basic reporting

Suggested improvements: what we learned from your findings that we did not know before? It is somewhat hard to find a take-home message, mainly from the Mb conclusions.

Experimental design

Measurements of EWL should be taken in experimental chambers of the same size for all the focal species since air turnover within the test container is different in different container sizes. The authors argued that it was not relevant since they did not compared rates of EWL among species. However, as it can be read, the manuscript discussion I centered in comparing species (of different habitat specialisation). The authors should also acknowledge more clearly the potential confounding effects of differential feeding states[A] (before their experiments) and dehydration (during their trials). The authors argued that they did not observe loss in mass that was great enough to suggest we had an issue with dehydration. However, does it mean “great enough”? A statistical test or reference should be provided to confirm this assumption since it is known that more dehydrated anurans decrease their rates of EWL[B], Tb[C], performance [D] and so forth [E].

A-Witters,L.R., Sievert,L., 2001.Feeding causes thermophily in the woodhouse´s toad (Bufo woodhousii). J.Therm.Biol.26,205–208.

B-Rodolfo C. O. Anderson, Rafael P. Bovo, Carlos E. Eismann, Amauri A. Menegario, and Denis V. Andrade, "Not Good, but Not All Bad: Dehydration Effects on Body Fluids, Organ Masses, and Water Flux through the Skin of Rhinella schneideri (Amphibia, Bufonidae)," Physiological and Biochemical Zoology 90, no. 3 (May/June 2017): 313-320.

C-Tracy C. R., Christian K. A., O'Connor M. P., & Tracy C. R. (1993). Behavioral thermoregulation by Bufo americanus: The importance of the hydric environment. Herpetologica, 49, 375–382.

D-Preest M. R., & Pough F. H. (2003). Effects of body temperature and hydration state on organismal performance of toads, Bufo americanus . Physiological and Biochemical Zoology, 76, 229–239.

E- Tracy, C. R., K. A. Christian, G. Betts, and C. R. Tracy. 2008. Body temperature and resistance to evaporative water loss in tropical Australian frogs. Comparative Biochemistry and Physiology A 150:102–108.

Validity of the findings

No more comments.

Additional comments

Abstract

Lines 21-23: It is hard to follow the rationale. This sentence should be rewritten.
What kind of process related to pronounced sexual dimorphism in anurans are influenced? Why more general factors like sex and body mass (Mb) would be related to a very specific factor like vapour pressure deficit (VPD)? Does not make sense for most of the readers.
VPD seems an operational variable and should be replaced by its theoretical variable, i.e. what VPD mean in practical terms.

If Abstract section has limit of words, the description of the three species with common names and etc. could be shorten. The most important is to highlight the habitat use of the three focal species.

You found that: 1-Tb, whole-animal EWL and whole-animal SMR increased with increasing Ta; 2-Tb increased with Mb, except in the more aquatic species; 3-whole-animal EWL was not affected by VPD, Mb or sex; 4-Whole-animal SMR increased with Mb in the more terrestrial species. From these findings, the conclusion was: “Although we found that Ta is very important in influencing key physiological processes” (that is ok), "the generality of Mb as an important factor should be taken in the context of species habitat specialisation."
What does you specifically mean? What does your work add to the knowledge gap? What is(are) the take home message(s) in practical terms? How could this link to the three physiological traits tested in the hypotheses? Be more specific and connect it with your findings in order to communicate the conclusion of your research better than it is. Even reading some of the authors’ replies, it is still hard to find a simple message from Mb effects on the physiological traits tested: “Lastly, looking specifically at the influence of Mb in our results, we are of the view that these species-specific Mb trends suggest complex Mb dependency that will not be easily solved statistically by lumping all, that is having mass instead of species will not be feasible.” Therefore, this is the job of the writers and not of the readers in highlighting “the generality of Mb as an important factor” that closes the Abstract and it seems one of the main conclusions of the manuscript.

Introduction

I commented on this in another round of review, however I could not find any predictions on expected effects of Ta on Tb at the final of the Introduction section. Although it is expected that both Ta and Tb would change in the same direction, this may not be true at certain conditions as, for instance, at high Ta’s when Tb decreases due to loss of water by evaporation through skin.

Line 104: positive?negative? relationship between water loss and activity patters.

Methods

Check for misspelling of “Respirometry“ throughout the text.

Anuran species were fed with different diet. Please explain why you used this treatment since different acclimation conditions may affect physiological traits in a differential way. Also, explain why you are confident that individuals did not get food differentially since there were more than one per plastic boxes at least for 2 species (there is no information on the third, Xenopus laevis). Different captivity conditions ("treatments") may imply in differences when analyzing physiological responses.
Author reply: Although this may be true, we tried to feed species the diet that they would encounter in nature. In addition to feeding individuals in groups as highlighted by the reviewer, after each experimental run, frogs were placed in individual holding cages and were fed before they were returned to the larger plastic terraria (holding cages), see lines 218- 223.
My reply: It is extraordinary to learn that in nature Xenopus feed on ox-heart. Authors were evasive in their reply to my comment and thus I suggest they acknowledge the potential of this effect on the physiological measurements on the Methods section. Also acknowledge your method for feeding animals not individually may have implied in different conditions of fasting since some individuals could have fed and other not.

Why did you used 2 hours for EWL measurements for all individuals independent of their body size? Do you have evidence that this time period did not caused dehydration, and then you include another factor other than temperature influencing the rates of EWL?
Authors reply: The percentage body mass loss in the 2 hr experimental period was (mean ±SE) 3. 2 1± 1.66 (Xenopus laevis), 3.91 ±3.84 % (Sclerophrys capensis and 5.2 ±2.85 % (Amietia delalandii). We have to assume the “other factor” referred to here, concerned VPD? Please see our response further above where we check that VPD did not confound our analyses as far as we could tell.
My reply: Basically, if some individuals or species become more dehydrated than others, this fact could imply in differential changes in Tb, EWL, and SMR, since the effect of dehydration would be a confounding factor and, thus, should be acknowledge in the text.

Thermoregulation and Tb relationships should not be discussed without referring their interplay with hydro-regulation. Even if you mention it in some lines below at EWL sub-section, the influence of body mass on Tb and metabolism cannot be decoupled from hydration state. If so, please provide rational explanation.
Authors reply: We always hydrated our frogs between experiments runs as indicated in the manuscript (see line 151-154) and even after experiments. During experiments, lasting two hours, we did not observe loss in mass that was great enough to suggest we had an issue with dehydration.
My reply: I meant that thermoregulation and Tb relationships should not be discussed without referring their interplay with hydro-regulation. It was not about the methods / experimental design.
By the way, what does you mean with “great enough”? A statistical test or reference should be provided to confirm this assumption.

Discussion:

Line 365: I have already commented on this: “Species with high rates of Rc…”
Rc is not a rate and I explained why in a revision before.

Lines 366-367: “and in this study we found that Tb is influenced by Mb in both S. capensis and A. delalandii.” Ok, but what does this mean in practical terms? Conclude that. One way would be to highlight what kind of “specialisation” (more terrestrial?) and also to discuss why this was found in your study. Actually, many parts of the discussion sections are like this: “Tracy et al.. showed this… Navas showed that… and we found this.” For example, lines 348-367. The reader expects that the authors do the job of interpreting the study findings and conclude what is being discussed. Please review all the Discussion section.

Line 374: Where you showed that you expected this? Actually, expectations for Tb are lacking at the end of the Introduction section.

Lines 383-387: Prates and Navas 2009 data/paper weakly relate to what is being discussed. I suggest you delete it or even replace by another one.

Conclusion:

Lines 434-436: Check again if this sentence makes sense or should be rewritten. In addition, it is hard to understand from where this suggestion comes from, since you did not link with any of your findings in the sentences before. Again, what we learned from your findings that we did not know before? You should focus on that.

---

## Round 0.4 · accepted · Accept

Thank you very much for your revision, which has, again, improved the manuscript. I do not see any issues precluding the publication, so I am happy to accept your interesting article!